# The Sacred River: State Ritual, Political Legitimacy, and Religious Practice of the Jidu in Imperial China

Teng Li 

College of Marxism, Shijiazhuang Tiedao University, Shijiazhuang 050043, China; liteng@stdu.edu.cn

**Abstract:** This paper focuses on the Jidu 濟瀆 (i.e., the Ji River 濟水), one of the four waterways (*sidu* 四瀆) in imperial China. Even though it vanished a long time ago, the Jidu had always been a part of the traditional Chinese ritual system of mountain- and water-directed state sacrifices. From the Western Han dynasty to the Qing dynasty, it continuously received regular state sacrifices. However, Western scholars have failed to notice it. Some modern Chinese and Japanese scholars have studied the development of the Jidu sacrifice, but its embodied political and religious significances for the state and local society were largely ignored. To remedy this neglect, I provide here, with new discoveries and conclusions, the first comprehensive study of the Jidu sacrifice in imperial China. Surrounding this coherent theme, this paper draws several original arguments from its four sections. The first section is a brief history of the state sacrifice to the Jidu. In the second section, I analyze the ideas of state authority, political legitimacy, religious belief, and cosmology, as these underlie the ritual performance concerning the Jidu. I argue that the Jidu was not only tightly associated with controlling water but was also a symbol and mechanism of political legitimacy. Relying on concrete official and local records, in the third section I further investigate the role that the Jidu God played in local society. I argue that after the Song dynasty, the Jidu God was transformed into a regional protector of local society and savior of local people in addition to an official water god. In the fourth section, I, for the first time, examine the interaction between the Jidu cult and other religious traditions including Daoism, Buddhism, and folk religion.

**Keywords:** sacred river; Jidu; state ritual system; political legitimacy; religious practice; imperial China

## 1. Introduction

An often-quoted sentence in the *Zuozhuan* 左傳 (*Zuo's Commentary*) reveals the significance of state sacrifice: "the two foremost matters of the state were those of sacrificial worship and war" 國之大事，在祀與戎 (Yang 1981, 8.861). State sacrifice was regarded as one of the events in traditional China most crucial to sustaining political legitimacy, ideological orthodoxy, bureaucracy, and social order. It was a huge and complicated ritual system. In this system, the sacrificial ritual concerning geographical features such as mountains, rivers, and seas were important components. From the earliest dynasties, mountains and rivers had been given political, cosmological meanings and were objects of divination. From the Western Han dynasty (206 BCE–8 CE) to the Northern Song dynasty (960–1126), the Chinese imperial courts gradually formed a standardized official sacred geographical system which consisted of five sacred peaks (*wuyue* 五岳), five strongholds (*wuzhen* 五鎮), four seas (*sihai* 四海), and four waterways (*sidu* 四瀆) (Jia 2021, pp. 1–12). For imperial courts, the integration of rituals of these mountain and water spirits was an effective way to manage the territory of the empire by connecting the state to local society.

This paper focuses on the Jidu 濟瀆 (i.e., the Ji River 濟水), one of the four waterways. The character "du" 瀆, according to the *Erya* 爾雅 (*Correct Words*), was interpreted: "the four waterways refer to the Yangzi River 長江, the Yellow River 黃河, the Huai River 淮河, and the Ji River. Each has its own source and flows to seas separately" 江、河、淮、濟為四

瀆。四瀆者，發源注海者也 (Guo and Xing 2000, 7.225). Nowadays, the Yangzi, the Yellow, and the Huai rivers still play crucial roles in China. Unlike the other three rivers, the Ji River has long disappeared. Nonetheless, a lot of places that contain "ji" 濟 (for example, Ji'nan 濟南, Jiyuan 濟源, and Jining 濟寧) prove its existence. In transmitted Chinese texts, the four waterways also manifested as the Jiangdu 江瀆, the Hedu 河瀆, the Huaidu 淮瀆, and the Jidu. The gods of the four waterways are accordingly called the Jiangdu God, the Hedu God, the Huaidu God, and the Jidu God. Additionally, they were often associated with directions: the Jiangdu with south, the Hedu with west, the Huaidu with east, and the Jidu with north, according to their locations.

Although vanished, the Jidu had always been a part of traditional Chinese ritual system of mountain- and water- directed state sacrifices. From the Western Han dynasty to the Qing dynasty (1644–1911), it continuously received regular state sacrifices. In imperial China, the Jidu sacrifice was not only a religious activity but also a political institution. Any comprehensive study of the Jidu should take this dual function into account. However, Western scholars have failed to notice the Jidu sacrifice. Some modern Chinese and Japanese scholars such as Yao Yongxia 姚永霞, Sakurai Satomi 櫻井智美, and Xiao Hongbing 肖紅兵 have studied the development of the Jidu sacrifice, but its embodied political and religious significances for the state and local society were largely ignored (Yao 2014; Sakurai 2014; Xiao and Li 2019). To remedy this neglect, I provide the first comprehensive study of the Jidu sacrifice. Surrounding this coherent theme, this paper comprises four sections. The first section is a brief history of the state sacrifice of the Jidu. The second section focuses on analysis of the ideas of state authority, political legitimacy, religious belief, and cosmology, as these underlie the ritual performance concerning the Jidu. Relying on concrete official and local records, in the third section I investigate the role that the Jidu God played in local society after the Song dynasty. In the fourth section, I, for the first time, examine the interaction between the Jidu cult and other religious traditions including Daoism, Buddhism, and folk religion.

## 2. A Brief History of the State Sacrifice to the Jidu

Inscriptions on oracle bones suggest that official and formal sacrifice to major rivers can be dated back to the Shang dynasty (ca. 1600–ca. 1046 BCE), and the Yellow River received most of the sacrifices. Because the names of the five sacred peaks and four waterways were mentioned in the three Confucian ritual classics, the *Liji* 禮記 (*Records of Ritual*), *Zhouli* 周禮 (*Ritual of Zhou*), and *Yili* 儀禮 (*Classic of Ritual*), some modern scholars have followed the traditional view that the composition of the four waterways was already completed in the Zhou dynasty. However, just as Jia Jinhua has pointed out, this view is unsubstantial because the date of compilation of these documents is questionable (Jia 2021, p. 3). By far, the earliest record of the sacrifice to the Jidu is found in the *Zuozhuan*:

> Ren, Su, Xuju, and Zhuanxu, whose surname is Feng, take duty of the sacrifice to the Taihao and Ji River. 任，宿，須句，顓臾，風姓也，實司大皞與有濟之祀. (Yang 1981, 5.391)

The record in the *Zuozhuan* only implies who (Ren, Su, Xuju, and Zhuanxu) takes the duty of sacrificing to the Jidu but fails to provide any details of the ritual. With Qin's unification, a new state sacrificial ritual system to integrate the mountain and river spirits was constructed. A detailed account of the process of this construction is preserved in the *Fengshanshu* 封禪書 (*Book of Feng and Shan Sacrifices*) in the *Shiji* 史記 (*Records of the Grand Historian*) by Sima Qian 司馬遷 (145–86 BCE). The texts read:

> When the First Emperor of Qin united the world, he instructed the officials in charge of sacrifice put into order the worship of Heaven and Earth, the famous mountains, the great rivers, and the other spirits that had customarily been honored in the past. According to this new arrangement there were five mountains and two rivers east of Xiao designated for sacrifice. The mountains were the Great Hall (that is, Mount Song), Mount Heng, Mount Tai, Mount Kuaiji and Mount Xiang. The two rivers were the Ji and the Huai. In the spring offerings of

dried meat and wine were made to ensure the fruitfulness of the year, and at the same time prayers were offered for the melting of the ice. In the autumn prayers were offered for the freezing of the ice, and in the winter prayers and sacrifices were offered to recompense the gods for their favor during the year. A cow and a calf were invariably used as sacrifice, but the sacrificial implements and the offerings of jade and silk differed with the time and place. 及秦并天下，令祠官所常奉天地名山大川鬼神可得而序也。於是自殽以東，名山五，大川祠二。曰太室。太室，嵩高也。恒山，泰山，會稽，湘山。水曰濟，曰淮。春以脯酒為歲祠，因泮凍，秋涸凍，冬塞禱祠。其牲用牛犢各一，牢具珪幣各異. (Sima 1963, 28.1371; Watson 1993, pp. 15–16)

Qin's reconstruction of the state sacrifice to the mountains and rivers was not only a compulsory means of strengthening imperium but also the first attempt at clarifying the order of the mountains and rivers that were already sacrificed to. Therefore, it is reasonable to speculate that the Jidu had already been sacrificed to by some regional states in the Spring and Autumn period and in the Qin dynasty (221–207 BCE), but there had not formed a standardized state sacrificial scheme of the four waterways.

In the early Western Han dynasty, the court basically followed the state sacrificial system and regulations of mountain and river spirits that were founded in the Qin dynasty. At the outset of his reign, Emperor Gaozu of Han 漢高祖 (r. 202–195 BCE) issued an edict to restore the state sacrifice in 201 BCE:

I hold the places of worship in the highest regard and deeply respect the sacrifices. Whenever the time comes for sacrifices to the Lord on High or for the worship of the mountains, rivers, or other spirits, let the ceremonies be performed in due season as they were in the past. 吾甚重祠而敬祭，今上帝之祭及山川諸神當祠者，各以其時禮祠之如故. (Sima 1963, 28.1378; Watson 1993, p. 19)

However, the real situation was that the kings of the princedom were powerful and held the authority of sacrificing to the mountains and rivers in their territories. Upon the collapse of the princedoms of the Huainan 淮南 and the Qi 齊, Emperor Wendi of Han 漢文帝 (r. 180–157 BCE) was able to resume his authority for sacrifice and again sent out the Grand Supplicant (*taizhu* 太祝) to perform the rituals. Like the First Emperor of Qin 秦始皇 (r. 247–221 BCE), Emperor Wudi of Han 漢武帝 (r. 141–87 BCE) showed the greatest reverence in the sacrificial rituals of *feng* and *shan* (*fengshan* 封禪) on Mount Tai 泰山 (in 110 BCE and 106 BCE). After Wudi's offering of the rituals of *feng* and *shan* on Mount Tai in 110 BCE, his second-half imperial tour included all the five sacred peaks and four waterways (Sima 1963, 28.1403).

The integration of the five sacred peaks and four waterways as a state sacrificial scheme was accomplished during the reign of Emperor Xuandi of Han 漢宣帝 (74–49 BCE). In 61 BCE, Xuandi issued an edict of rearranging great mountains and rivers by making new adjustments to the offerings, as the *Hanshu* 漢書 (*Han History*) records:

"The Yangzi River is the biggest of hundreds. But until now it had no temple to sacrifice to it. Therefore I (the emperor) command the officials in charge of sacrifice to take the sacrificial rituals into account and regard them as anniversary ceremonies. Sacrifice to the rivers of Yangzi and Luo at each of the four seasons to pray for harvests in the whole country." Since then, all the five sacred peaks and four waterways have had regular sacrifices. The eastern sacred peak (Mount Tai) is sacrificed to in Bo; the central sacred peak (Mount Taishi) is sacrificed to in Songgao; the southern sacred peak (Mount Qian) is sacrificed to in Qian; the northern sacred peak (Mount Chang) is sacrificed to in Quyang. The Hedu is sacrificed to in Linjin; the Jiangdu is sacrificed to in Jiangdu; the Huaidu is sacrificed to in Pingshi; the Jidu is sacrificed to in Linyi. All of them above should be sacrificed to by officials dispatched by the court with tally. Only Mount Tai and the Hedu are sacrificed to five times annually, while the Jiangdu is sacrificed to four times annually, and all the rest are prayed to once and sacrificed to three

times annually. "夫江海，百川之大者也，今闕焉無祠。其令祠官以禮為歲事，以四時祠江海雒水，祈為天下豐年焉。"自是五嶽、四瀆皆有常禮。東嶽泰山於博，中嶽泰室於嵩高，南嶽灊山於灊，西嶽華山於華陰，北嶽常山於上曲陽，河於臨晉，江於江都，淮於平氏，濟於臨邑界中，皆使者持節侍祠。唯泰山與河歲五祠，江水四，餘皆一禱而三祠云. (Ban 1964, 25.1249)

This Han edict not only clearly regulated the place (Linyi 臨邑, present day Dezhou 德州, Shandong), frequency (three times a year), and people (imperial commissioners) of sacrifice to the Jidu, but also confirmed a uniform practice of regular sacrifice to the five sacred peaks and four waterways.

During the period of division, one of the most significant developments was the formation of the five rites (*wuli* 五禮). The state ritual of the Jidu and other three waterways was a part of the auspicious rites (*jili* 吉禮). Many powers in this period had tried to continue or restore the practice of sacrificing to the five sacred peaks and four waterways. For example, in 221, Emperor Wendi of Wei 魏文帝 (r. 220–226) issued an edict to "sacrifice to the five sacred peaks and four waterways" 初祀五嶽四瀆 (Du 1988, 46.1281). In 399, Emperor Daowu of Northern Wei 道武帝 (r. 386–409) hosted the state sacrifices at the northern suburb of the capital city Pingcheng 平城 (present day Datong, Shanxi). The five sacred peaks and four waterways were sacrificed to symbolically: "the five peaks and other famous mountains were sacrificed to in the inner altar; the four waterways and other great rivers were sacrificed to in the outer altar" 五岳名山在中壝内，四瀆大川於外壝内 (Du 1988, 45.1260). In 418, the Northern Wei court erected a Temple of Five Sacred Peaks and Four Waterways (*Wuyue sidu miao* 五岳四瀆廟) on the south bank of the Sanggan River 桑乾水 near the capital city. According to the chapters of sacrifices in official histories during the period of division, almost all the powers had put the five sacred peaks and four waterways in the list of officially worshipped mountains and rivers, even though some were not located in their territories. It suggests that these mountains and rivers were not only geographical landscapes but also symbols of state unification and political legitimacy.

In 582, during the reign of Emperor Wendi of Sui 隋文帝 (r. 581–604), the Jidu Temple (*Jidu miao* 濟瀆廟) was established. Two years later, the Jiyuan district 濟源縣 was established. The name "Jiyuan" literally means "source of the Jidu". Before the Sui dynasty, the directors of the temples of four waterways were the Grand Supplicant and shamans, who were religious officials. However, starting with the Sui dynasty, the manipulation of the Jidu Temple was taken over by administrative officials. The *Suishu* 隋書 (*Sui History*) records:

> The magistrates of the five sacred peaks, four waterways, and Mount Wu . . . are ranked deputy eighth grade. 五岳、四瀆、吳山等令 . . . . . . 為視從八品. (Wei 1973, 28.790)

The Tang court carried on the instalment of the Waterway Magistrate (*duling* 瀆令), but the bureaucratic ranking descended from the deputy eighth grade to ninth grade. According to the *Xin Tangshu* 新唐書 (*New Tang History*), a complete operation team of the temple, then, consisted of thirty-four persons, comprising one Waterway Magistrate, three Supplication Scribes (*zhushi* 祝史), and thirty Court Gentlemen for Fasting (*zhailang* 齋郎) (Ouyang and Song 1975, 49.1321). In the Jidu Temple during the Tang dynasty, the magistrate of Jidu was the honest protector and operator of state sacrifice. His works varied from preparing sacrificial material to hosting the whole ceremony. As an official with a bureaucratic grade, he was the representative of the emperor and court.

The conferment of titles on the Jidu began during the Tang dynasty (Zhu 2007, 2022). In 747, each of the four waterways was conferred an official title. The Jidu was entitled Duke of Pure Source (*Qingyuan gong* 清源公); the other three waterways were also granted titles: the Jiangdu Duke of Grand Source (*Guangyuan gong* 廣源公), the Hedu Duke of Efficacious Source (*Lingyuan gong* 靈源公), and the Huaidu Duke of Long Source (*Changyuan gong* 長源公) (Liu 1975, 24.934; Zhu 2022, p. 2). In 751, Emperor Xuanzong of Tang 唐玄宗 (r. 712–756) issued an edict to dispatch some high-ranking officials to offer sacrifice to the sacred peaks, strongholds, waterways, and seas in each local temple (ibid.).

The tendency of granting titles to the four waterways originating from the High Tang continued in the Song dynasty. In 1040, all the four waterways were promoted from Duke (*gong* 公) to King (*wang* 王) and, accordingly, the Jidu was granted the title King of Pure Source (*Qingyuan wang* 清源王) (Toqto'a 1977, 102.2488). In 1125, the Jidu was given a new title: King of Loyal and Protective Pure Source (*Qingyuan zhonghu wang* 清源忠護王). The term "zhonghu" 忠護 literally means "loyal and protective". Official documents failed to record this title, but I find it in the *Jiyuanxian Zhi* 濟源縣志 (*Jiyuan District Gazetteer*), which was compiled in the Qing dynasty. The local gazetteer kept the record of a Song edict originally carved on a stone. The stone is now missing. It narrates a story that tells how the Jidu God manifested his power to bring heavy rain to quell an invasion of bandits from a neighboring county, and the court therefore granted him this new title (Xiao 1976, 16.673)[1].

When the Yuan empire was founded, the Mongolian court imitated the Tang's and Song's continually granting of titles on the four waterways (Ma 2011). According to the *Yuanshi* 元史 (*Yuan History*), in 1291 the titles of each of the four waterways gained two more characters and the Jidu was granted the title Savior King of Pure Source (*Qingyuan shanji wang* 清源善濟王) (Song 1976, 76.1900).

In 1370, Emperor Taizu of Ming 明太祖 (r. 1368–1398) issued an edict to justify the liturgical reform of state rituals of the gods of sacred peaks, strongholds, seas, and waterways. One effect of this edict was to remove the titles that had been granted by previous regimes. This edict was carved on steles and sent to temples. One of them still stands in the Jidu Temple now; that is, "Daming zhaozhi bei" 大明詔旨碑 (Stele of the Imperial Edict of the Great Ming). The texts read:

> The way of governing must be rooted in the rites. The granting of titles to the sacred peaks, strongholds, seas, and waterways was from the Tang and Song. For them, the brilliant and numinous material forces were concentrated to form their spirits, and who received the mandate from the High God. How can anything be added to them by investiture or the bestowal of honorific titles by the ruling house? In profanation of the rites, nothing could be more inappropriate than this. Now we follow the ancient regulations and deprive the titles which were granted in previous dynasties ... The four waterways are called "God of the Eastern Waterway Great Huai," "God of the Southern Waterway Great Jiang," and "God of the Western Waterway Great He," "God of the Northern Waterway Great Ji." 為治之道，必本於禮。嶽鎮海瀆之封，起自唐、宋。夫英靈之氣，萃而為神，必受命於上帝，豈國家封號所可加？瀆禮不經，莫此為甚。今依古定制，並去前代所封名號 ... ... 四瀆稱東瀆大淮之神，南瀆大江之神，西瀆大河之神，北瀆大濟之神. (Zhang 1974, 49.1284; Feng 2012, pp. 7–9)

For Emperor Taizu, the official granting of titles on these earthly deities violated Confucian values. Therefore, he believed these titles must be stripped. This was a part of his religious reform. When Emperor Taizu ascended the throne, he soon launched many measures to re-evaluate and manage religions in general. He tried to involve Buddhism, Daoism, and folk religion in an officially controlled system. By reconstructing a religious system, state orthodoxy was established. Despite the fact that the official granting of titles was abolished, other aspects of the ritual, such as the date, place, procedures, and so on, remain unchanged.

By the Qing dynasty, in 1723 Emperor Yongzheng 雍正 (r. 1722–1735) issued an edict to confer a new title to the Jidu. The Qing official histories failed to record the full name. Fortunately, it was mentioned in the 1723 imperial edict which was etched on a stele in the Jidu Temple. As the inscription records, the full title is "God of Forever Beneficial Northern Waterway Great Ji" (*Beidu yonghui daji zhishen* 北瀆永惠大濟之神) (Yao 2014, pp. 69–70). Even though it faced complicated external environment and internal crises, the late Qing court did not abolish the state ritual of the Jidu. State sacrifices were regularly performed in the local temples of the sacred peaks, strongholds, seas, and waterways. Stele inscriptions in the Jidu Temple show that, during the reign of Emperor Guangxu 光緒 (r. 1875–1908), the court still dispatched officials to offer the regular sacrifice to the Ji River. Upon the

collapse of the Qing dynasty, the two-thousand-year-old state ritual ultimately vanished as a practical concern.

Previous scholarship on Chinese religion attached much more attention on concrete cults and ritual practice in the local level. However, as shown earlier, the state sacrifice to the Jidu was granted political and symbolic meanings and therefore played an important role in imperial China. This section, by providing a historical account, not only clarifies the development of state sacrifice to the Jidu but also constructs a broader historical context for the following sections.

**3. The Structure and Significance of the State Ritual of the Jidu**

The Tang was a vital phase in the formation of a standardized imperial ritual of the Jidu. Starting with the Tang dynasty, the state ritual of the Jidu officially became a part of the auspicious rites and was leveled in the medium sacrifice (*zhongsi* 中祀). The Tang books of rites, particularly the *Datang Kaiyuanli* 大唐開元禮 (*Kaiyuan Ritual of the Great Tang*), standardized the place, date, procedures, participants, sacrificial offerings, words of prayer, and other elements (Xiao 2000). This sacrificial paradigm was inherited by successive dynasties. According to the *Datang Kaiyuanli*, there are four sites of sacrificing to the Jidu: (1) suburbs of the capital city, including the northern suburb *(beijiao* 北郊) and southern suburb (*nanjiao* 南郊); (2) the palace; (3) Mount Sheshou 社首山; (4) the Jidu Temple (Xiao 2000, 36.201-202, 62.321-328, 64.338-345, 65.345-347, 66.347-349, 67.349-351). Now I introduce them in turn.

First, in a complete north suburban sacrifice in Tang China, the four waterways were regarded as gods subordinate (*congsi* 從祀) to the two first-leveled gods listed in the major sacrifice (*dasi* 大祀), the Grand Deity of Earth (*huangdiqi* 皇地祇) and Divine Land (*shenzhou* 神州). The annual ceremony of the Tang north suburban sacrifice was held on the day of summer solstice (*xiazhi* 夏至) in the square altar (*fangqiu* 方丘). The four waterways were also sacrificed to in the round altar (*yuanqiu* 圜丘) of the southern suburb. This sacrifice is called La Sacrifice (*laji* 臘祭), which was held on the eighth day of the last month (*laba* 臘八). Unlike the north suburban sacrifice, the south suburban sacrifice was offered for Hundreds of Gods (*baishen* 百神), including the Jidu God (Wechsler 1985, pp. 118–20). Whether in the square altar or round altar, the statute of the Jidu God was arranged at the northern part of the outer wall (*waiwei* 外壝). In addition to the regular sacrifice, when there was a prolonged drought or continuous rain, the court would dispatch officials in charge of sacrifice to make a sacrifice to the Jidu to generate or to stop rain, mostly in the northern suburb.

Second, the Jidu was sacrificed to in the palace. According to the *Datang Kaiyuanli*, a remote sacrifice to the mountains and rivers (not limited to the sacred peaks, strongholds, and waterways) was offered immediately when the emperor's imperial carriage returned from an imperial tour of inspection (Xiao 2000, 62.321).

Third, the *Datang Kaiyuanli* records a detailed sacrificial ritual of *shan* (*shanli* 禪禮) at Mount Sheshou. The ritual was held randomly as a part of the sacrificial rituals of feng and shan. The Jidu was sacrificed to during the ceremony of the sacrificial ritual of shan at the Mount Sheshou (ibid., 64.338–345).

Fourth, from the Tang dynasty, the four waterways were sacrificed to once a year in their own local temples: the Jiangdu in Yizhou 益州 (present day Chengdu, Sichuan), the Hedu in Tongzhou 同州 (present day Dali 大荔, Shaanxi), the Huaidu in Tangzhou 唐州 (present day Tongbai 桐柏, Henan), and the Jidu in Luozhou 洛州 (present day Jiyuan, Henan). The date of performing the ritual was called "the days of greeting the seasonal *qi* in the five suburbs" (*wujiao ying qi ri* 五郊迎氣日). According to their directions and the theory of five phases (*wuxing* 五行), the Jiangdu is sacrificed to on the day of the start of summer (*lixia* 立夏), the Hedu on the day of the start of autumn (*liqiu* 立秋), the Huaidu on the day of the start of spring (*lichun* 立春), and the Jidu on the day of the start of winter (*lidong* 立冬).

*3.1. The Structure of the State Ritual of the Jidu Held in the Jidu Temple*

Of all the sacrifices to the Jidu, none was as important as those held in the Jidu Temple. As mentioned above, the state ritual of the Jidu leveled the medium sacrifice. However, only that was held in the Jidu Temple was qualified as medium sacrifice. As for the suburban sacrifices at the square altar and round altar, its level varied with the center god. The Tang court first promulgated a detailed ritual code for offering sacrifices in the Jidu Temple, which is mainly preserved in three Tang texts: *juan* 36 of the *Datang Kaiyuanli*, no. 72 of Ritual (*li* 禮) of the *Tongdian* 通典 (*Compendium of Comprehensive Institutions*), and a mid-Tang stele inscription written by Zhang Xi 張洗 (fl. late eighth century to early ninth century), who then was the District Governor of Jiyuan (*jiyuanxian yin* 濟源縣尹; Xiao 2000, 36.201-202; Du 1988, 112.2897-2903). The name of this stele inscription is "Jidumiao Beihaitan jipinbei" 濟瀆廟北海壇祭品碑 (Stele of the Sacrificial Offerings for the North Sea Altar and the Jidu Temple), which records a complete ritual of sacrifice to the Jidu and the North Sea (*beihai* 北海) in 797 (Wang 1985, 103.1733). These texts portray a colorful picture of the standardized annually regular sacrifice to the Jidu in the Jidu Temple.

According to these texts, there were six phases of the ritual. The six phases are outlined below:

1. Preparation for the ritual. According to the *Datang Kaiyuanli*, ritual officials were required to take part in the ritual of abstinence (*zhai* 齋) to purify themselves before the ceremony began. The complete ritual lasted five days, including a three-day partial abstinence (*sanzhai* 散齋) and a two-day complete abstinence (*zhizhai* 致齋). During the three-day partial abstinence, the ritual officials could deal with routine administrative affairs as usual in the daytime and stay at home at night. However, that which was thought to be polluted should be forbidden, such as mourning for dead, visiting sick people, signing criminal documents, having sex, and so forth. The two-day complete abstinence was even stricter, and everything was forbidden but sacrificial matters. Ritual officials must stay at the temple and rehearse the ceremony. The Tang institution of abstinence was derived from the *Liji*. As written in the *Liji*, the purpose of abstinence was to purify the ritual officials' heart-mind and body: "the abstinence is achieved when the highest degree of refined intelligence is reached. After this it is possible to enter into communion with the spirits" 齊者精明之至也，然後可以交於神明也[2] (Zheng and Kong 2000, 49.1575).

2. Preparation of the sacred space. After the ritual of abstinence, the Jidu Magistrate cleansed the temple and dug a deep pit (*kan* 埳) in the north one day before the ceremony. The pit was used for the burial of sacrificial offerings. An altar with many steps was built in the pit. According to the *Datang Kaiyuanli* and *Tongdian*, the Jidu Magistrate was also obliged to arrange the positions of the ritual participants and major sacrificial wine and food vessels. Three Supplication Scribes stood at the southeast side of the altar, while facing toward the northwest; the Hymn Singer (*zanchangzhe* 讚唱者) stood at the southwest of the presenters; the Priest (*jiguan*祭官) stood at the northwest side of the altar.

3. Cooking food for the god. On the eve of the ceremony day, the Court Gentlemen for Fasting slaughtered the sacrificial animals and put their blood and fur into the wooden vessels (*dou* 豆), and then placed them in the kitchen. At dawn on the ceremony day, the chef cooked these animals in the kitchen. A full banquet (*tailao* 太牢) was offered, including an ox, a pig, and a sheep. The color of ritual animals was black. In addition to animals, more than twenty kinds of dishes and four kinds of wines were offered (Wang 2021, p. 6). The offering of food and drink not only expressed sincere and deep respect for the gods but also showed the prosperity of an agrarian empire.

4. Getting ready for the ritual. As the officially appointed director in charge of temple affairs, the Jidu Magistrate must be prepared before the start of the formal ceremony in the morning. According to the *Datang Kaiyuanli*, the Jidu Magistrate led the Supplication Scribers and the Court Gentlemen for Fasting to stand to the east of the altar. The statue of the Jidu God was then raised up in the middle of the altar. Four bottles

(*zun* 樽) of wine, two pieces of jade with bottoms (*lianggui youdi* 两圭有邸), one piece of black silk, and a prayer tablet (*zhuban* 祝版) were arranged in their correct positions. When the Hymn Singer was ready, the Supplication Scribes and bearers of wine and food approached the altar, waiting for the ritual of Three Offerings (*sanxian* 三獻).

5.　The Three Offerings. In the morning, the ritual began. The Receptionist (*zanlizhe* 贊禮者) guided the ritual officials to wait outside the altar. About half an hour later, these officials were led to their stations. The Hymn Singer intoned: "kowtow twice" (*zaibai* 再拜)". All the participants kneeled to kowtow. Then the Receptionist went to the left side of the First Supplicant (*chuxianguan* 初獻官) and instructed him to perform the First Offering (*chuxian* 初獻). Afterwards, the Receptionist guided the First Supplicant to enter the altar and to stand in front of the statue of the Jidu God. Then, the jade and silk were presented. After that, the First Supplicant returned to his station and the sacrificial food was presented. The First Supplicant was led to wash hands and clean winecups, then proceeded to the statue of the Jidu God. He kneeled again, took up the winecup of sweet wine, and drained it. After that, he descended from the altar. The Supplication Scribes ascended the altar holding the prayer tablet, kneeled, and read the prayer words on the tablet. Once finished, the tablet was put under the statue of the Jidu God, and the First Supplicant drank a cup of pure wine. The Receptionist then guided the Second Supplicant (*yaxianguan* 亞獻官) to wash hands and winecups. The Second Supplicant then ascended to the altar from the east side and was led to the front of the statue of the Jidu God. He then kneeled, faced north, and drained the cup of wine. Then another cup of pure wine wad delivered to him. He drank it and returned the winecup. The Second Offering (*yaxian* 亞獻) was completed. Once finished, the Receptionist guided the Second Supplicant to his station. The Third Offering (*zhongxian* 終獻) was offered by the Third Supplicant (*zhongxianguan* 終獻官), following the same procedure as that of the Second Supplicant.

6.　Sinking the silk and burning the prayer tablet. After the Three Offerings, the Jidu Magistrate and the Court Gentlemen for Fasting sank the silk and ritual animal blood. Then the Hymn Singer sang: "ritual ends". All participants kneeled and kowtowed for the last time. They then returned to the place of abstinence (*zhaisuo* 齋所). The prayer tablet was burned as the final phase.

In the sacrificial ceremony, the most important participants were the Three Supplicants. As recorded in the "Jidumiao Beihaitan jipinbei", the First Supplicant was the Regional Inspector, the Second Supplicant was Zhang Xi himself, the District Governor of Jiyuan, and the Third Supplicant was the Vice Magistrate of Jiyuan. According to the *Datang Kaiyuanli* and *Tongdian*, the numbers of sacrificial vessels, ritual animals, offerings, dishes of food, and wine strictly followed a hierarchy. Sacrificial vessels presented to the Jidu included six bottles, ten bamboo-made vessels (*bian* 籩), ten wooden vessels, two round bowls (*gui* 簋), two square bowls (*fu* 簠), and three big plates (*zu* 俎). The ritual animals included one ox, one pig, and one sheep. The offerings consisted of two pieces of jade and some black silk. More than twenty dishes of food were offered (Xiao 2000, 36.201). In addition, three kinds of wine were provided: one bottle of sweet wine (*liqi* 醴齊), one bottle of rice wine (*angqi* 盎齊), and one bottle of pure wine (*qingjiu* 清酒).

### 3.2. Communicating with the God: The Significance of the Ritual

In imperial China, especially after Confucianism had been officially accepted as the state ideology in the Han dynasty, the notions of ritual, kingship, power, state religion, and political legitimacy were found to be closely interdependent, and these were all engaged in the sacrificial ceremony. By extracting and developing some essential issues from the Confucian classics (especially from the three ritual classics, the *Zhouli*, *Liji* and *Yili*), the newly standardized rituals were regarded as the most effective means of connecting mortals and gods, or terrestrial and celestial realms. The purpose of the ritual was to establish some direct connections with the celestial realm. Therefore, during the process of the state ritual

of the Jidu, all the concrete procedures and sacrificial items in use were given religious and political significance.

On the day of sacrifice, the whole process of the state ritual of the Jidu was performed at the altar in a deep pit. According to the Tang ritual code, the gods of the four waterways were categorized in the group of Earthly Deity (*diqi* 地祇); thus, the shape of the sacrificial altar was square. This follows from the primary principle in ancient Chinese cosmology: "the heaven is round and the earth is square" (*tianyuan difang* 天圓地方). Unlike the disordered folk rituals in the local community, state ritual created an orderly, sacred place that was independent of outer geography. The altar was designed with many steps (*bi* 陛), which were seen as a symbol distinguishing the human and spiritual worlds. The supplicants ascended the steps from the bottom to the peak of the altar, implying their transcendence of the terrestrial world to the celestial world. To fit themselves for attendance in the celestial realm, the supplicants must participate in the ritual of abstinence to purify themselves physically and mentally and to show sincerity.

The various ritual vessels, as Wu Hung argues, not only have their practical function as implements for food and wine, but also embodies ritual codes and political power as ceremonial paraphernalia for specific ritual purposes (Wu 1993, p. 24). On the one hand, the number of the sacrificial vessels suggests a hierarchy of the gods. On the other hand, the sacrificial offerings also had special religious meanings. Food, including the meat, fish, cereals, and vegetables, were media of communication between gods and humans. Even in China today, sacrificial food is never wasted. Chinese people believe that they will receive good fortune if they eat sacrificial food. This belief implies that the food which is used to feast gods has been "delivered" to the spiritual world and "returned" to the human world with a little remnant power and good fortune from the gods.

Wine was another way of feasting gods and deities in the Tang ritual. As mentioned earlier, there were three kinds of wine provided in the sacrificial ritual to the Jidu. Different wine was offered for different purposes: the sweet wine and the rice wine were used to feast the god, while the pure wine was prepared for the supplicants. This distinction, in Roel Sterckx's words, is "securing a balance between the entertainment of spirits with food and drink and the desire for convivial celebration by ritual participants" (Sterckx 2011, p. 98).

Jade and silk were offered to the gods and deities in most of the imperial rituals. In ancient Chinese philosophy, jade was viewed as one of the purest natural products. Silk, produced by silkworms, was regarded as a gift from nature. Therefore, the ancient Chinese believed that they were able to connect heaven, earth, and human, viewing them as symbols of the "unity of human and heaven" (*tianrenheyi* 天人合一).

In the state ritual of the Jidu, the tablet was a knot directly binding human emperors and spiritual gods. When making a tablet, one must follow strict standards of material, length, width, and height (Ouyang and Song 1975, 12.332). Prayer words were carved on the tablet, and the tablet was offered to the Jidu God at the end of the ceremony. It was used to deliver information from the emperor to the god. According to the *Datang Kaiyuanli*, each waterway had its own prayer words. The prayer words for the Jidu God read:

> For the Northern Waterway Great Ji: "You have a pure source, fertilize the far and near regions, flow four kinds of energies, and discipline the area. Offer sacrifice to you in the winter according to the state rites. 北瀆大濟云：維神泉源清潔，浸被遐邇，播通四氣，作紀一方，玄冬肇節，聿修典制. (Xiao 2000, 36.202)

As mentioned earlier, the first official title granted to the Jidu was Duke of Pure Source. From the Tang dynasty, the most significant feature of the Jidu was "pure", as the prayer words summarized. In the ritual, the prayer words were not only used to inform the Jidu God that the offerings were well prepared, but also to propitiate him with highly praised characters.

According to the *Datang Kaiyuanli*, the jade and silk were sunk in the river after the last offering by the Jidu Magistrate and the Court Gentlemen for Fasting (ibid.). The symbolic purpose was to deliver the sacrificial offerings to the god in the water. The last step of the

whole ritual was to burn the prayer tablet at the place of abstinence (ibid.). The tablet was seen as a medium of delivering information from the human world to the heavenly realm, as the smoke produced by burning the tablet was thought to reach Heaven.

In the context of the Tang state ritual code, the state ritual of the Jidu was two-fold: suburban and regular sacrifice in the Jidu Temple. In general, the latter was a simplified version of the former in respect to ritual procedure and basic sacrificial elements. The ritual in the Jidu Temple had more functions and meanings than did suburban counterparts. All sacrificial offerings were imbued with religious meaning. Political legitimacy and the emperor's authority were also emphasized through the ritual. Although emperors never participated in local sacrifice, prayer tablets sealed by them were taken as an effective substitute for their presence.

In this section, I, for the first time, clarify the six phases of the state ritual of Jidu that were held in the Jidu Temple by relying on three Tang texts, and then analyze the ideas of state authority, political legitimacy, religious belief, and cosmology, as these underlie the ritual performance concerning the Jidu. I argue that the Jidu was not only tightly associated with controlling water but was also a symbol and mechanism of political legitimacy. The next section will take up the legendary stories of the Jidu God looking at the role he played in local religious life.

## 4. Divinity on the Stele: The Jidu God in Local Society

Located in Jiyuan, Henan Province, the Jidu Temple impresses tourists for its magnificent ancient halls and pavilions. With over 30 buildings constructed from the Song to Qing, the Jidu Temple is the only surviving and largely intact architectural structure connected to sacrifice to the four waterways. Historically, it was the site of sacrifice to the Jidu and North Sea. As mentioned earlier, the Jidu Temple was established in the Sui dynasty. Unfortunately, there are no Sui and Tang architectural structures remaining except for a broken wall. Extant structures were mostly built in the Ming and Qing dynasties, while the earliest can be dated back to the Song dynasty. Like the majority of traditional Chinese temples, it is composed of rectilinear complexes of building all cardinally orientated to the south (Wheatley 1971, pp. 147–58).

Inside the Jidu Temple, there are more than 160 inscribed steles (from the Tang dynasty to Republican period). In addition, the *Jiyuanxian Zhi* and *Xu Jiyuanxian Zhi* 續濟源縣志 (A Supplement of Jiyuan District Gazetteer) also preserve some lost stele inscriptions (Xiao 1976; He 2013). In general, according to the contents, preserved stele inscriptions in the Jidu Temple and local records can be divided into five types:

1.  Imperially composed invocations (*yuzhi jiwen/zhuwen* 御制祭文/祝文). These official edicts were addressed to the Jidu God on behalf of the emperor. Most of them were carved in the Ming and Qing dynasties.
2.  Records of the ritual of tossing dragons and tablets. There are at least six stele inscriptions picturing this ritual. Interestingly, all of them are dated in the Yuan dynasty.
3.  Records of the restoration of the Jidu Temple by missionary or local officials.
4.  Legendary and efficacious stories of the Jidu God.
5.  Steles commemorating the merit and virtue (*gongdebei* 功德碑) of local people for their donation to the Jidu Temple.

According to these stele inscriptions and local records, since the Tang dynasty, the Jidu God was not only regarded as the most reliable official water god but also as symbol of political legitimacy, regional protector of local society, and savior of local people. I shall now discuss these divine functions in turn.

Making rain was the most important divine function of the Jidu God. Almost all the Ming–Qing imperially composed invocations concerning the Jidu God were issued as a prayer for rain. For example, in 1527, because of a prolonged drought, Emperor Shizong of Ming 明世宗 (r. 1522–1566) ordered the major local officials of Henan to sacrifice to the Jidu God for rain. Here is the inscription of this event:

The state will turn to offering sacrifice to the gods and ghosts when disasters and famines happen. This follows the rite. From last winter to this spring, it did not snow or rain at all in north and south banks of the Yellow River . . . . . . the wheats were not fully grown, which led to a poor harvest. Seeding was particularly hard, and people are still starving. Officials had made sacrifice to all gods. . . . God of the pure Ji River which originates from Mount Wangwu fertilizes the lands and benefits the people with great merits. Sacrifices to you in past years always gained good results. Now, I (the emperor) reverently ordered the official [five characters missing] to offer the silk to you. You enjoy sacrifice in this quarter and it is your duty to solicitude and protect the district. Hope you silently operate your transformative power to make great rains, so that [three characters missing] the people could rely on you. 國有兇荒索鬼神而祭之，禮也。顧惟大河南北，自冬俱春雪雨全無，⋯ ⋯ □麥未成既歉，自穀播種尤艱，下民嗷嗷，有司用□，靡神不舉 ⋯ ⋯ 惟神清濟之流，發源王屋，利澤生民，功莫大焉。往歲事禱，恆獲嘉應，茲特敬恭，竭誠□官□告□□□幣昭假於神。享祀一方，殄恤捍禦，神之職也。尚其默運化機，沛以升雨，俾侍□□□民庶攸賴. ([Yao 2014], p. 220)

According to the inscriptions, the Ming emperor tried to sustain an emperor–minister relation for the sake of ordering the Jidu God to bring rain. After the Jidu God was bestowed noble titles, the emperor was obliged to issue an imperial edict to order him to control water. Words such as "it is your duty 神之職也" and "to make great rains 沛以升雨" frequently appeared in imperial edicts. The emperor, in the name of the Son of Heaven, was bound up with the system of sacrifice and maintenance of dynastic continuity and in his person was located the vital link between the divine cosmos and humanity ([Campany 2009], p. 199).

In imperial China, the Jidu God was also used to promote imperial indoctrination, which secured him a political function. A Song stele inscription titled "Chongxiu Jidumiao bei" 重脩濟瀆廟碑 (Inscription of Restoration of the Jidu Temple) clearly points out:

The [Jidu] God follows the mandate of heaven and silently promotes imperial indoctrination. The God takes responsibility for fertilizing lands and controlling rain. It is also the God's power that makes a good harvest. The god resonates to our benevolent government and enjoys our sacrifices, so that the people have a peaceful and rich life. 惟神上應天命，陰助皇化，膏澤調順，神之職也；多稼豐登，神之力也。感我德政，歆我祀事，故生民泰然. ([Xiao 1976], 16.665-666)

This stele was erected in 973 and was written by the early Song politician Lu Duoxun 盧多遜 (934–985). In Lu's words, the Jidu God was not only a water god in charge of favorable weather and good harvest but also a god who helped to promote imperial indoctrination and political legitimacy. The purpose of Lu's inscription was to demonstrate the orthodoxy of the Song after the conquest of the state of Southern Han 南漢.

In the Song dynasty, the Jidu God was regarded as a protector who quelled banditry and secured local people by controlling rain. As shown earlier, it won him an officially granted title of "King of Loyal and Protective Pure Source". Here is the story:

The King of Pure Source in the Jidu Temple takes advantage of his great power to benefit the local people. When the bandits of the neighboring county prepared to invade the border of Jiyuan County, local people ran to pray to you, the Jidu God. Thunder and rain arrived very quickly. The Qin River immediately had an inaccessible stronghold. Orderly banners suddenly appeared at the south bank, which looked like a strict troop. It seemed that all the bandits were deprived of their soul, and they fled away. People in communities kept undisturbed and then celebrated the victory. A memorial was presented to the emperor. The good efficacy was so obvious that the emperor sighed with pleasance. In order to commemorating this event, the emperor ordered a great title be bestowed to the god for repaying his miraculous help and consoling the people's hearts. The god descends down to enjoy the sacrifice and the people along the river reply on him. The god shall be granted King of Loyal and Protective Pure Source. 濟瀆廟清

源王，利澤溥博，陰福吾氏屬者。寇發鄰郡，將犯縣境，邑人奔走禱于爾大神，雷雨迅興。沁河有湯池之險，旌旗猋列南岸，象羽林之嚴。賊徒裭魄以咸奔，閭里按堵而相慶。奏函來上，休應昭然，嘉歎不忘，宜崇美號，庶苔靈眖，式慰民心，來格來歆，一方水賴，可特封：清源忠護王. (Xiao 1976, 16.673; Yao 2014, p. 208)

In this story, the Jidu God was regarded as a territory protector by the Jiyuan people. It reveals a trend that official water gods such as the Jidu God descended from the state ritual code to local society from the Song dynasty onward.

In the Yuan dynasty, the Jidu God was offered sacrifices by the Mongolian court for the sake of exterminating a locust plague. According to the stele inscription of "Huangtaizi Yanwang sixiang beiji" 皇太子燕王嗣香碑記 (Record of Pilgrimage of the Crown Prince of Yan), in 1272 the crown prince of Yan, Borjigin Zhenjin 孛兒只斤·真金 (1243–1285) sent Daoist priests to practice the Great Ritual Offerings to the All-Embracing Heaven (*luotian dajiao* 羅天大醮) in the Jidu Temple (Chen 1988, p. 1102). To borrow the great supernatural power to fight against the serious plague of locusts from the Jidu God, this ritual was exclusively offered to him. The locust plague usually occurred because of severe drought. Therefore, the real intention of offering sacrifice to the Jidu God was to pray for rain to relieve the drought.

According to these stories, I argue that since the Song dynasty, the Jidu God had been transformed into a regional protector and savior of local people in addition to an official water god. Through a continuous stream of legendary, miraculous intervention, including relieving drought, bringing rain, quelling flood, fending off bandits, and subduing disasters, the Jidu God renewed his bond to the local communities and secured people's devotion.

## 5. The Jidu Cult in Other Religious Traditions

From the Tang dynasty there were obvious interactions of official sacrifice and institutionalized religions such as Daoism and Buddhism. Daoist priests transformed the cults of the sacred peaks, strongholds, seas, and waterways by involving them in the Daoist pantheon and rituals. With the spread of Daoism, these transformed rituals were gradually accepted by the common people and merged with folk belief. They were also absorbed into Buddhist ritual. This section examines religious beliefs and practices of the Jidu God beyond the state ritual code by looking at it in multiple religious dimensions.

In medieval China, the most flourishing Daoist ritual regarding the Jidu was tossing the dragons and tablets (*toulongjian* 投龍簡). This ritual, developed in the fifth century, was for the sake of praying for blessing and eradicating disasters.[3] According to the ritual site, there were normally two methods of performing the ritual: burial and sinking. Daoist priests buried the written prayer on the tablets with the green silk threads (*qingsi* 青絲), golden dragons (*jinlong* 金龍), and golden rings (*jinniu* 金紐) in the mountains, grotto heavens (*dongtian* 洞天), and blessed places (*fudi* 福地). In the rituals that were performed at the waterways and lakes, Daoist priests threw these ritual objects into the water and sank them.

Extant stele inscriptions suggest that the Daoist Ritual of Tossing Dragons and Tablets began to be performed in the Jidu Temple during Empress Wu's 武后 reign (690–705). In 691, only one year after Empress Wu's ascension to the throne, she sent a Daoist priest, Ma Yuanzhen 馬元貞 (fl. eighth century), who was abbot of the Jintai Abbey 金臺觀 in the capital city Chang'an 長安 (present day Xi'an, Shaanxi), to perform the rituals at the five sacred peaks and four waterways for the purpose of obtaining merits (Lei 2009, pp. 153–66). From 691 to 692, two of Ma Yuanzhen's disciples (Yang Jingchu 楊景初 and Guo Xiyuan 郭希元), and two officials from the court (Yang Junshang 楊君尚 and Ouyang Zhicong 歐陽智琮) performed the rituals. With an obvious intention of political propaganda designed to secure legitimacy, the rituals were performed for fulfilling Empress Wu's political ambitions. Therefore, as "Fengxianguan Laojun shixiang bei" 奉仙觀老君石像碑 (Inscription of the Stone Statue of the Elderly Lord in the Fengxian Abbey) records, after Ma Yuanzhen and his disciples erected a statue of the Grand Supreme Elderly Lord (*Taishang Laojun* 太上老君),

there were auspicious signs: a crane flew around and auspicious clouds were manifested (Chen 1988, p. 80).

There were three kinds of tablets: the mountain tablet (*shanjian* 山簡), water tablet (*shuijian* 水簡), and earth tablet (*tujian* 土簡). The water tablets were tossed in the auspicious springs, seas, and the four waterways (Lingbao yujian 1988, p. 0333). The only extant water tablet thrown in the Jidu was discovered at the Jidu Temple in 2003. It is a piece of rectangular jade tablet with a few lines of prayer. The text reads:

> The great Song son and subject of Heaven (one character missing) . . . twenty-one persons opened (two characters missing) the Golden Register Fasting Ceremony . . . Throwing the golden dragons and jade tablets into the Water Bureau, wish the Gods, the three officials, the Nine Emperors of Water Bureau . . . present to the Nine Heavens. Cautiously reach the golden dragon station of the Water Bureau dispatch the information. On the wushen day of the fourth month in the first year of the Xining (1068). 大宋嗣天子臣□ . . . . . . 三七人開啓□天□金籙道場 . . . . . . 水府投送金龍玉簡，願神願仙，三元同存，九府水帝 . . . . . . 奏，上聞九天。謹詣，水府金龍驛傳。熙寧元年太歲戊申四月. (Yao 2014, p. 55)

From the fragmentary inscriptions, we can tell that this piece of jade tablet was tossed into the Jidu after a Golden Register Fasting Ceremony (*jinluzhaiyi* 金籙齋儀). According to the date, the ritual was performed just after Emperor Shenzong of Song 宋神宗 (r. 1067–1085) ascended to throne. The emperor tried to declare his emperorship and to present this information to Heaven through the ritual.

After the Song dynasty, emperors of the Jin and Yuan dynasties continued the Daoist Ritual of Tossing Dragons and Tablets. As the stele inscriptions in the Jidu Temple show, it was performed even more frequently. During this period, Confucian officials and Daoist priests were frequently sent by the court to practice the rituals (Ma 2011). However, from the Ming dynasty on, as the decreasing number of stele inscriptions concerning the ritual in the Jidu Temple suggests, the Daoist Ritual of Tossing Dragons and Tablets was gradually decreased by the imperial courts.

The Jidu God was also absorbed in the Chinese Buddhist Water–Land Ritual (*shuilu fahui* 水陸法會) from the late Tang dynasty. According to the ritual text *Fajie Shengfan Shuilu Shenghui Xiuzhai Yigui* (1975) 法界聖凡水陸勝會修齋儀軌 (*The Fasting Rite of the Most Excellent Ceremony in Which All Enlightened and Unenlightened Beings of Land and Water Share a Great Meal to Aid Liberation*), the Jidu God and other three gods of waterways were invited to the inner hall during the first night. In the ritual, they were called Source Dukes of the Four Waterways (*Siduyuan gong* 四瀆源公) (X. 1497.3a). As mentioned earlier, this title was conferred by the Tang court, which reveals a medieval framework of the Buddhist Water–Land Ritual.

For the imperial courts, the gods of four waterways were regarded as quasi-officials in the divine bureaucratic system of the celestial realm such as the God of Earth (*tudishen* 土地神) and the City God (*chengchuangshen* 城隍神). However, for the common people they were defined as water gods by their function of controlling waters. "In China, divinity is a responsibility like a public function: the title endures nut those who hold it succeed one another . . . . . . They are functionary gods who receive a position, who lose it, who are promoted or demoted" (Maspero 1981, p. 87). Therefore, the image of the Jidu God was reshaped in folklore and popular literature.

According to one of the most significant and influential works of folk religious literature, *Sanjiao Yuanliu Soushen Daquan* 三教源流搜神大全 (The Comprehensive Collection of Investigations into the Divinities of the Three Doctrines since Their Origin), each of the four waterways had its correlative god (Qin 2012, p. 60). Here, the gods of four waterways were categorized in the group of Confucian Gods (*rujiaoshen* 儒教神). They shared a title of "Gods of the Four Waterways (*sidushen* 四瀆神)". All were real historical figures who devoted themselves to the state. More importantly, they were all ministers with high bureaucratic rankings. The Hedu God was Chen Ping 陳平 (d. 178 BCE), the Western Han Counselor-in-chief. The Jiangdu God was Qu Yuan 屈原 (340–278 BCE), one of the

most famous politicians and poets in Chinese history. The Huaidu God was Pei Du 裴度 (765–839), the mid-Tang Grand Councilor.

The Jidu God was Wu Zixu 伍子胥 (d. 484 BCE). He was a general and politician of the Wu State 吳國 during the Spring and Autumn period, who was famous for his loyalty. According to the *Shiji*, not long after Wu Zixu was forced to commit suicide, the Wu people began to worship him (Sima 1963, 66.2180). The local people of the Wu State built him a shrine not only in honor of his contribution but in sympathy with his sufferings. In the *Sanjiao Yuanliu Soushen Daquan*, Wu Zixu was also worshipped as the God of Tide (*chaoshen* 潮神). The cult of Wu Zixu was particularly popular in the Jiangzhe area 江浙地區. However, he had nothing to do with the Jidu. He neither took the office in the ancient Ji River area nor left a legendary story there. It is likely that the compilers of the folk religious literature deliberately fabricated a connection of Wu Zixu and the Jidu. However, the Jidu God had his political significance. Worship of these prestigious officials, as John Shryock stated, "has been encouraged by the government, since it holds up examples of good men for public emulation and encourage virtue by keeping alive the memory of great deed. Doubtless there is also the feeling that benefit may accrue to the worshipers from the increased power of the hero in the next world" (Shryock 1931, p. 45).

## 6. Conclusions

In this paper, focusing on the coherent theme of comprehensively understanding the rich implications of the Jidu sacrifice in imperial China, I have examined the history, structure, and significance of the state sacrificial ritual of the Jidu, as well as the Jidu cult in local society and other religious traditions. Four original, major arguments and conclusions are drawn from the examination.

First, I indicate that in imperial China the Jidu was not only tightly associated with controlling water but was also a symbol and mechanism of political legitimacy by analyzing the ideas of state authority, political legitimacy, religious belief, and cosmology, as these underlie the ritual performance concerning the Jidu. The state ritual of the Jidu held in the Jidu Temple was the most important Jidu sacrifice during imperial times.

Second, this paper provides the first detailed analysis of the six phases of the state ritual of the Jidu held in the Jidu Temple, which is a new discovery. Performing the ritual was thought to be an effective means of connecting mortals and gods, or terrestrial and celestial realms. All the ritual procedures and sacrificial offerings were imbued with religious and political meaning.

Third, the Jidu Temple acted as a node connecting the state and local society. Political legitimacy and the emperor's authority were preached through regular sacrifice and the participation of imperial commissioners, local bureaucrats, ritual specialists, and people. The stele inscriptions in the Jidu Temple suggest that the imperial courts had always tried to impose official water gods such as the Jidu God in local society by holding frequent and regular sacrificial ceremonies. The Jidu God was therefore worshiped in local society. From the Song dynasty, in addition to acting as an official water god in the state ritual code, the Jidu God was transformed into a territory protector by local people and often manifested his divine figure when there was a need.

Fourth, I, for the first time, examine the religious beliefs and practices of the Jidu God beyond the state ritual code by looking at it in multiple religious dimensions. From the late Tang dynasty, the Jidu and the Jidu God began to be associated with various religious traditions. They became involved in the Daoist Ritual of Tossing Dragons and Tablets and the Buddhist Water–Land Ritual. Particularly when Daoist priests were trusted by the Jurchen and Mongolian rulers during the Jin and Yuan dynasties, they often undertook imperial missions to offer sacrifices to the Jidu. From the Song dynasty, the Jidu God was more widely acknowledged and attracted more believers. The prestigious bureaucrat Wu Zixu, who conformed to orthodox Confucian values, became the Jidu God in folk religious literature. This not only implies that the Jidu cult was widely shared by the common people, but also reveals a process of reproduction from high culture to low culture.

Overall, as this paper has demonstrated, in imperial China the Jidu and the Jidu God meant many things to different people. For the imperial courts, the Jidu not only satisfied the need to control water in an agrarian empire but was also a political symbol and mechanism of imperial legitimacy. For the Daoist priest, the Jidu was a sacred site to practice the Daoist Ritual of Tossing Dragons and Tablets. For the Buddhist clergy, the Jidu God acted as one of the protective deities in particular rituals. For the people in local society, the Jidu God was thought to be a territory protector. Thus, although the Jidu sacrifice has been largely ignored in previous scholarship, it was crucial to imperial China's politics and played an important part in the history of Chinese religion. It deserves further and deeper explorations. **Funding:** This research received no external funding.

**Data Availability Statement:** Not applicable.

**Conflicts of Interest:** The author declares no conflict of interest.

## Notes

[1] I shall discuss this story in the fourth section.

[2] The Chinese character *qi* 齊 is often used in the sense of *zhai* 齋 (abstinence) in traditional texts.

[3] More precisely, just as Édouard Émmannuel Chavannes (1865–1918) observed, the most flourishing period of practicing the ritual of tossing dragons and tablets was from the 7th century to the 14th century. See (Chavannes 1919, pp. 53–220). According to the objects to be tossed, the *tou longjian* literally means tossing (or casting) dragons and tablets. See (Wang 2012, p. 51). In this ritual, both golden dragons and tablets will be cast in the end. They are two different objects, but some scholars mistake it as "tossing the dragon tablets". For example, see (Huang 2012, p. 234; Raz 2010, p. 421). Chavannes first noticed this Daoist ritual and its religious functions in medieval China in the early 20th century. However, only recently have Chinese and Japanese scholars begun to pay attention to it again. See (Chavannes 1919, pp. 53–220; Kamitsuka 1992, pp. 126–34; Zhou 1999, pp. 91–109; Lei 2004, pp. 73–80; Lei 2009, pp. 153–66; Liu 2007, pp. 235–70; Zhang 2007, pp. 27–32; Huang 2012, pp. 234–39; Xie 2018, pp. 228–46; Yi 2018, pp. 132–73; Lü 2019, pp. 91–101).

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
