# Peer review of "The Sacred River: State Ritual, Political Legitimacy, and Religious Practice of the Jidu in Imperial China"

_religions, doi:10.3390/rel13060507_

Round 1
Reviewer 1 Report
There is a need for copyediting.
The author might want to reconsider some of the English translations of Chinese terms.
The article tends to be very descriptive and textual without engaging any major debates in the field, and engaging the earlier literature. It will be good to have more of that in the conclusion.
Author Response
Point 1: There is a need for copyediting.
Response:
The previous version of my paper was reviewed by a UK professor and proofread by a retired Cambridge University expert. This new revised version again has been proofread by a native English expert.
Point 2: The author might want to reconsider some of the English translations of Chinese terms.
Response:
The previous version of my paper was reviewed by a renowned UK professor, and a famous Chinese scholar has also helped check the translation and interpretation of cited primary sources. I hope the reviewer could indicate any concrete errors in the translation so that I am able to correct them.
Point 3: The article tends to be very descriptive and textual without engaging any major debates in the field and engaging the earlier literature. It will be good to have more of that in the conclusion.
Response: As I have suggested in the abstract, this paper is the first comprehensive study of the Jidu sacrifice in all languages. Western scholars have overlooked the Jidu, while the Chinese and Japanese scholars have only studied the development of the Jidu sacrifice. There have never been any debates in previous scholarship. I have cited major Chinese, Japanese, and English studies in the first version of my paper, and I have added more in the revised version. My paper focuses on the central theme of Jidu sacrifice and draws four original arguments through critical analysis of primary and secondary sources. I have now revised the parts of abstract, introduction, and conclusions to make my critical arguments more clear.
Reviewer 2 Report
This is well-argued article based on thorough textual research, with good use of etymological reasoning. The article is technically well executed and based on state-of-the-art secondary literature.
Author Response
Point 1: This is well-argued article based on thorough textual research, with good use of etymological reasoning. The article is technically well executed and based on state-of-the-art secondary literature.
Response 1: I really appreciate your comments. Thanks!
Reviewer 3 Report
A key problem with this article is that it does not put forward any coherent argument. There is much interesting material, but the author does not do very much with it. For example, all the records going down through the dynasties do not really tell us much about the reality of sacrifices to the Ji River. The material is delivered in quite a raw form without much reflection on what it might mean. Similarly, the long descriptions of various rituals in this article do not advance any argument. For example, on pp.5-6, the author makes four points about rituals, but does not tell us what we should learn from them. There is no argument. This also occurs in later description of ritual. It is not enough to say what happened in a sacrifice; the author needs to build this material into an argument of some kind. In fact, some key points are buried. For example, in the first paragraph of page 4, the key point is in the last sentence. The paragraph should really be reframed and that point emphasized.
Another problem is that much of this material is not properly contextualized. As a result, its meaning is not apparent. For example, the conferring of titles on the four waterways under the Tang was part of a broader trend of granting titles to regional gods that started under the Tang, probably prompted by the actions of Wu Zetian. I think there is a recent book in Chinese about this, but I cannot recall the title. The author might also want to consider the reordering of sacrifices under Xuanzong at more length with reference to Sima Chengzhen. Robert Hymes, Sue Takashi and others have written about the granting of official titles to regional gods under the Song and what that was about.
Similarly, some of the translations of texts are given without any analysis. So what are the translations for? For example, the translation on p.5 is not discussed. Its meaning is quite complex and could be used to develop the article in various ways. The author should not simply present a long rich text without actually using it.
In fact, the whole article reads like it was written in Chinese and translated to English; this is not necessarily a problem in itself, but in this case it means that the article does not follow the norms of western Sinological writing, working instead in a Chinese mode. There are assumptions about the accuracy of the sources used and these sources are not placed in context or subject to proper examination. For example, Shiji and Hanshu of course offer idealized views of some events; they can be used as sources, but with some care. These issues are not so much an issue in Chinese academia, but they are problems for an article in English, which is subject to different demands. It is actually not acceptable just to transfer material from one realm to the other for publication without transforming it. This demonstrates an ignorance of academic discourse in English.
This article has very few references to works published in English and other western languages; this is quite a serious failing, since there are many sources that would benefit analysis. For example, the first chapters of James Robson‘s The Religious Landscape of the Southern Sacred Peak might provide some useful background for the state of research when it was published. Furthermore, I would note that Japanese authors are only used in Chinese translation, meaning the wealth of material in Japanese potentially relevant to this work is also not accessed. For Tang sacrifices on p.6, the author could refer to the work of Lei Wen.
It might also be worth rethinking the references to Chinese classical sources. Ban 1964 is an odd way to refer to Hanshu. Most academic papers would refer to the work rather than the author for such works.
The article needs to be framed around an argument rather than just a list of interesting materials without context. To develop a clearer argument, some of the material under 3.2 could perhaps be placed at the centre of the work alongside the material about the history of Jidu in order to determine its changing position in state and local sacrifices, possibly with the aim of examining the tensions between the two.
Finally, the English expression is occasionally problematic and some translations are awkward.
That said, I did enjoy reading the article, but it needs to be totally reframed and the author needs to consider what the article is trying to argue. There is much potential here but this potential requires more than just a ten-day rewrite.
Author Response
Point 1: A key problem with this article is that it does not put forward any coherent argument.
Response:
Most sophisticated academic articles have more than one argument. My paper focuses on the coherent theme of Jidu sacrifice, and surrounding this theme I draw four major arguments through critical analysis of primary and secondary sources. 1. The Jidu not only tightly associated with controlling water but was also a symbol and mechanism of political legitimacy. 2. Performing the state sacrificial ritual to the Jidu was thought to be an effective means of connecting mortals and gods, or terrestrial and celestial realms. 3. After the Song dynasty the Jidu God was transformed into a regional protector of local society and savior of local people in addition to an official water god. 4. After the Tang dynasty, the Jidu cult interacted with other religious traditions including Daoism, Buddhism, and popular religion. In this new version, I have revised the parts of abstract, introduction, and conclusions to make my arguments more clear, and I have also more clearly summarized my arguments at the end of each section.
Point 2: There is much interesting material, but the author does not do very much with it. For example, all the records going down through the dynasties do not really tell us much about the reality of sacrifices to the Ji River. The material is delivered in quite a raw form without much reflection on what it might mean.
Response: In section 2, I use stele inscriptions preserved in the Jidu temple to analyze the actual performance of the sacrificial ritual held in the Jidu temple in the Tang dynasty. I select a representative inscription from dozens of steles, and analyze the six phases of sacrifice ritual described in the inscription. This is the most solid and detailed example for offering the picture of the reality of sacrifice to the Ji River. In this revised version, I have added more analyses on citations of primary sources (for example, line 132-134, 264-271, 498-502).
Point 3: Similarly, the long descriptions of various rituals in this article do not advance any argument. For example, on pp.5-6, the author makes four points about rituals, but does not tell us what we should learn from them. There is no argument. This also occurs in later description of ritual. It is not enough to say what happened in a sacrifice; the author needs to build this material into an argument of some kind.
Response: My purpose of the long descriptions of various rituals is to locate the Jidu sacrifice in the Tang ritual code. This is very crucial. The theme of this paper is to comprehensively study the Jidu sacrifice. We need to know the role the Jidu sacrifice played in the state sacrificial system. More importantly, on pp. 5-6, I for the first time indicate that there were four sites of sacrificing the Jidu, which is a new discovery. I would like to indicate again that my paper presents four original, major arguments, which have never been talked about by other scholars.
Point 4: In fact, some key points are buried. For example, in the first paragraph of page 4, the key point is in the last sentence. The paragraph should really be reframed and that point emphasized.
Response: Thanks for the suggestion. I have reframed this paragraph and emphasized my argument at the end of this section.
Point 5: Another problem is that much of this material is not properly contextualized. As a result, its meaning is not apparent. For example, the conferring of titles on the four waterways under the Tang was part of a broader trend of granting titles to regional gods that started under the Tang, probably prompted by the actions of Wu Zetian. I think there is a recent book in Chinese about this, but I cannot recall the title.
Response: It is incorrect to say “the conferring of titles on the four waterways under the Tang was part of a broader trend of granting titles to regional gods.” The four waterways, along with the five sacred peaks, four strongholds (became five from the Song), and four seas were not regional gods since the Han dynasty. They formed a state sacrificial system to mountain and water spirits and therefore were “official gods” or “state gods,” and their temples were state facilities. The conferment was in fact a return to Confucian classics such as the Ritual of Zhou and Record of Ritual, in which the five sacred peaks and four waterways are described as equal to dukes and regional lords. On the contrary, as I reveal in this paper, the Jidu God was transformed from a stat god to a regional god after the Song dynasty. The Jidu God is a good example of how official gods descended to local cults. This is one of my original and major arguments.
As far as I am concerned, I have not heard about the book mentioned in the comments, could the reviewer name it?
Point 6: The author might also want to consider the reordering of sacrifices under Xuanzong at more length with reference to Sima Chengzhen. Robert Hymes, Sue Takashi and others have written about the granting of official titles to regional gods under the Song and what that was about.
Response: As recorded in the Jiu Tangshu and other sources, Sima Chengzhen’s request of reordering the sacrifices under Tang Xuanzong only concerned the five sacred peaks and two other mountains (Mt. Tiantai and Mt. Wangwu). It had nothing to do with the Jidu sacrifice.
The works of James Watson (Standardizing the Gods), Robert Hymes (Way and Byway), Sue Takashi (唐宋期における祠廟の廟額・封号の下賜について), Valerie Hansen (Changing God in Medieval China), Hamashima Atsutosh (総管信仰), Barend ter Haar (Guan Yu), etc., have indeed studied the granting of official titles to regional gods under the Song and after. But as mentioned above, this kind of conferment was opposite to the Jidu God that transformed from an official god to a regional god. Therefore, these studies are unrelated to my paper.
Point 7: Similarly, some of the translations of texts are given without any analysis. So what are the translations for? For example, the translation on p.5 is not discussed. Its meaning is quite complex and could be used to develop the article in various ways. The author should not simply present a long rich text without actually using it.
Response: Thanks for this suggestion. I have added more detailed analyses on all citations from primary sources.
Point 8: In fact, the whole article reads like it was written in Chinese and translated to English; this is not necessarily a problem in itself, but in this case it means that the article does not follow the norms of western Sinological writing, working instead in a Chinese mode. There are assumptions about the accuracy of the sources used and these sources are not placed in context or subject to proper examination. For example, Shiji and Hanshu of course offer idealized views of some events; they can be used as sources, but with some care. These issues are not so much an issue in Chinese academia, but they are problems for an article in English, which is subject to different demands. It is actually not acceptable just to transfer material from one realm to the other for publication without transforming it. This demonstrates an ignorance of academic discourse in English.
Response: In fact, this paper is an enhanced version of the essence of my PhD dissertation, which was directly written in English. My supervisor, a distinguished UK professor, had reviewed and revised it for several times. We discussed almost all the sentences of my dissertation. In addition, the dissertation was also proofread by a retired Cambridge University expert.
As far as I am concerned, in recent decades western sinologists do not question the authenticity of Shiji and Hanshu when citing them, except some particular parts or passages.
Point 9: This article has very few references to works published in English and other western languages; this is quite a serious failing, since there are many sources that would benefit analysis. For example, the first chapters of James Robson‘s The Religious Landscape of the Southern Sacred Peak might provide some useful background for the state of research when it was published.
Response: James Robson’s book is very good, but it focuses on the Nanyue, the southern sacred peak. In Chapter 1, Robson provided a historical-religious (esp. Buddhism and Daoism) background of Nanyue. Although Nanyue is one of the five sacred peaks, it is strange he never mentioned the sacrificial system of five sacred peaks, five strongholds, four waterways, and four seas. Therefore, the book is unrelated to my theme of Jidu sacrifice and its inclusion in the state ritual system. In fact, Jidu sacrifice was totally ignored by western scholars, and my paper is the first study of this theme in western languages.
I have referenced all the related, major secondary sources in writing this paper. In this new version, I have added a number of indirectly related studies.
Point 10: Furthermore, I would note that Japanese authors are only used in Chinese translation, meaning the wealth of material in Japanese potentially relevant to this work is also not accessed. For Tang sacrifices on p.6, the author could refer to the work of Lei Wen.
Response: The only Japanese who has paid attention to the Jidu sacrifice is Sakurai Satomi. Her article cited in my paper was directly written in Chinese and published in a Chinese journal (元代的岳瀆祭祀). Another of her works related to the Jidu is a paper about a stele inscription in the Jidu Temple (元至元九年「皇太子燕王嗣香碑」をめぐって). Although it does not actually contribute to my argument, I have added it in this new version. Other than Sakurai’s works, there are no Japanese works about the Jidu sacrifice.
I have added Lei Wen’s book Jiaomiao zhiwai 郊廟之外 and his another work.
Point 11: It might also be worth rethinking the references to Chinese classical sources. Ban 1964 is an odd way to refer to Hanshu. Most academic papers would refer to the work rather than the author for such works.
Response: This is the house style of Religions. I strictly follow it.
Point 12: The article needs to be framed around an argument rather than just a list of interesting materials without context. To develop a clearer argument, some of the material under 3.2 could perhaps be placed at the centre of the work alongside the material about the history of Jidu in order to determine its changing position in state and local sacrifices, possibly with the aim of examining the tensions between the two.
Response: As mentioned above, I do draw four major arguments surrounding the central theme of Jidu sacrifice, and all these arguments are original and well documented.
Point 13: Finally, the English expression is occasionally problematic and some translations are awkward.
Response: Both the previous version and this revised version have been proofread by native English experts.
Reviewer 4 Report
The paper makes a significant contribution to an important yet so far little studied ritual aspect of Chinese religion. The author did a good job in presenting the history and the multiple aspects of the jidu god and relevant ritual. Th paper makes very good use of primary sources and presents them coherently and clearly.
Minor suggestions: 1) p. 5. The author may want to connect the state ritual of jidu worship more concretely with the scholarly discussion on the early Song and early Ming imperial reordering of Chinese religious landscape. There are a lot of English literature on this, and the author may want to engage them a bit more. 2) p. 10. The jidu temple impressed tourists: even though the author mentioned it before, it helps if the author repeats the information on where the temple is actually located to avoid confusion on the sudden mention of "tourists" here. 3) p. 14: the Confucian gods, or rujiao shen: is the "rujiao shen" a generic term used in primary sources, or is it the author's own wording? Since the concept of "rujiao" is closely related to the early 20th century reconstruction of Chinese religions, the use of this term needs to be clarified.
Author Response
Response to Reviewer 4
Point 1: The paper makes a significant contribution to an important yet so far little studied ritual aspect of Chinese religion. The author did a good job in presenting the history and the multiple aspects of the jidu god an relevant ritual. The paper makes very good use of primary sources and presents them coherently and clearly.
Response: I really appreciate your comments. Thanks!
Point 2: p. 5. The author may want to connect the state ritual of jidu worship more concretely with the scholarly discussion on the early Song and early Ming imperial reordering of Chinese religious landscape. There are a lot of English literature on this, and the author may want to engage them a bit more.
Response: The first section is just a brief history of the state sacrifice to the Jidu. It constructs a broader historical context for following sections. In p. 6, I analyze why Emperor Ming Taizu stripped the titles of the mountain and water spirits and indicated that it was a part of the early Ming religious reform. On the other hand, since this paper exclusively focuses on the Jidu, it would be a bit redundant to connect the state ritual of Jidu more concretely with discussions on the early Song and early Ming imperial reordering of Chinese religious landscape.
Point 3: p. 10. The jidu temple impressed tourists: even though the author mentioned it before, it helps if the author repeats the information on where the temple is actually located to avoid confusion on the sudden mention of “tourists” here.
Response: Thanks for this suggestion. I have added the information on where the temple is located.
Point 4: p.14: the Confucian gods, or rujiao shen: is the "rujiao shen" a generic term used in primary sources, or is it the author's own wording? Since the concept of "rujiao" is closely related to the early 20th century reconstruction of Chinese religions, the use of this term needs to be clarified.
Response: The “rujiao shen” is a generic term used in the Sanjiao Yuanliu Soushen Daquan, which was compiled in the Ming dynasty. In fact, the term of “rujiao shen” in this popular religious text mainly refers to the gods who had a Confucian origin. It is different from the concept of “rujiao” in the context of early 20th century reconstruction of Chinese religions.
Reviewer 5 Report
This paper on the Jidu is firmly contextualized, coherently argued, and well written. I suggest accepting it for publication, and have only a few minor suggestions that the author may want to take into consideration when revising the paper:
1) The abstract should be revised in a more succinct but cogent way. Currently, a high proportion of it is overlapped with the main body of the paper.
2) It seems that the phrase 五郊迎氣日 on Page 7 should be translated as “the days of greeting the seasonal qi in the five suburbs” rather than “the five days of greeting the seasonal qi in the five suburbs.”
3) It seems better to translate the term 按堵 on Page 12 as “kept undisturbed” rather than “stopped them”.
4) In conclusion, the author may want to place his argument about the Jidu in a broader context, especially in the system consisting of the five sacred peaks (wuyue 五岳), five strongholds (wuzhen 五鎮), four seas (sihai 四海), and four waterways (sidu 四瀆). Hopefully, we can expect to have a better understanding of the Jidu by comparing and contrasting it with those similar religio-political phenomena.
Author Response
Response to Reviewer 5
Point 1: This paper on the Jidu is firmly contextualized, coherently argued, and well written. I suggest accepting it for publication and have only a few minor suggestions that the author may want to take into consideration when revising the paper.
Response: I really appreciate your comments. Thanks!
Point 2: The abstract should be revised in a more succinct but cogent way. Currently, a high proportion of it is overlapped with the main body of the paper.
Response: Thanks for this suggestion. I have revised the abstract to make it more succinct.
Point 3: It seems that the phrase 五郊迎氣日 on Page 7 should be translated as “the days of greeting the seasonal qi in the five suburbs” rather than “the five days of greeting the seasonal qi in the five suburbs.”
Response: Thanks for your suggestion. I have revised it.
Point 4: It seems better to translate the term 按堵 on Page 12 as “kept undisturbed” rather than “stopped them”.
Response: Thanks for your suggestion. Your translation is better. I have revised it.
Point 5: In conclusion, the author may want to place his argument about the Jidu in a broader context, especially in the system consisting of the five sacred peaks (wuyue 五岳), five strongholds (wuzhen 五鎮), four seas (sihai 四海), and four waterways (sidu 四瀆). Hopefully, we can expect to have a better understanding of the Jidu by comparing and contrasting it with those similar religio-political phenomena.
Response: This paper belongs to the special issue “Traditional Chinese State Ritual System of Sacrifice to Mountain and Water Spirits”. The first paper in this special issue, written by Prof. Jia Jinhua, has already introduced the formation of the ritual system of five sacred peaks, five strongholds, four seas, and four waterways and elaborated their embodied religious-political conceptions. Other scholars who have contributed papers to this special issue have studied specific sea, river, or mountain respectively. In the “Introduction” of my paper, I do use the first two paragraphs to place Jidu in the context of this ritual system. I think this should be sufficient, because the reader can get a better understanding of the whole system by reading other papers included in this special issue.
Round 2
Reviewer 1 Report
I feel that the introduction and conclusion still does not really engage the field of Chinese religion. I don't really see substantive changes there. It does not connect to the wider literature in both Chinese and English (and perhaps Japanese too), not just for the Jidu but also to such state rituals as well as the relationship between Chinese religion, society, and state. It emphasizes the importance of the Jidu but does not show how the excellent empirical work done actually engages the wider field of Chinese religion or the connections between religion/society/environment, for which there must be ample work in Chinese, Japanese, and English. For example, even discussing and engaging more of the works listed under references would be interesting.
The author also does not deal enough with research design, questions, hypotheses and methods as highlighted in your evaluation form. He or she basically says the previous three authors who have written on the Jidu did not cover certain aspects without explain what their works were actually about. The author then embarks on his/her narrative and presentation of findings. This is also an important part of the contextualization which I feel is missing.
For the English, if you are going to do the copyediting for them (or have someone else do so), it should be no issue. Otherwise, the author should get the help of another native expert to read through the manuscript or, at least, he or she should tell the existing one that there is no need to retain mistakes and stylistic issues to reflect the style of the author. I spotted various grammatical mistakes and stylistic issues in the revised version too. The person helping the author(s) needs to do a more consistent job of helping him/her/them clean up these mistakes. The new version is certainly better than the earlier one. It is clear that the Cambridge professors whom they mentioned in the responses were not reading the paper for language. Of course, I must emphasize that I am not from Cambridge. Unfortunately, my highlights are all signed so I did not share them, and will not share them now. Nevertheless, the editors can give it a close read to see what I mean. If they see no problem with the language, then my opinion will be irrelevant I guess.
I think the editors can decide on whether the authors need to do the first of what I highlighted, for the introduction and conclusion. I had put major revision because I feel that the issues highlighted for the introduction and conclusion really needs to be addressed. Nevertheless, it is just my opinion. I think the editors can decide if this is a case of "accept after minor revision" or "reconsider after major revision".
I feel that this article deserves publication definitely on the basis of the hard work put into the research and the very interesting data presented. It really depends on the journal editors and board to decide if the contextualization and engagement with the wider field of Chinese religion is needed for the articles in the journal.
Author Response
Response to Reviewer 1
Point 1: I feel that the introduction and conclusion still does not really engage the field of Chinese religion. I don’t really see substantive changes there. It does not connect to the wider literature in both Chinese and English (and perhaps Japanese too), not just for the Jidu but also to such state rituals as well as the relationship between Chinese religion, society, and state.
Response: The study of Jidu sacrifice and Jidu God cult itself is an important part of Chinese religion study. I have referenced over seventy Chinese, English, Japanese, and French literature to write this paper. These sources cover studies of state ritual, Buddhism, Daoism, popular religion, and the relationship between Chinese religion, society, and state. I engage with all these works throughout the paper. This reviewer’s comment, “it does not connect to the wider literature in both Chinese and English (and perhaps Japanese too),” is untrue and unfair. Has this reviewer ever read through my paper or did he/she only read the introduction and conclusion?
Point 2: It emphasizes the importance of the Jidu but does not show how the excellent empirical work done actually engages the wider field of Chinese religion or the connections between religion/society/environment, for which there must be ample work in Chinese, Japanese, and English. For example, even discussing and engaging more of the works listed under references would be interesting.
Response: There have been thousands of works on Chinese religion and the connections between religion/society/environment, but most of them are unrelated to the central arguments of this paper. This paper focuses on the Jidu, one of the four waterways, and it is the first study of Jidu sacrifice in Western languages and first comprehensive study in all languages. There have never been any debates on this topic. Because of limited space, there is no need to engage the wider field of Chinese religion. This is just a paper, not a book with over 300 pages.
Point 3: The author also does not deal enough with research design, questions, hypotheses and methods as highlighted in your evaluation form.
Response: To remedy the neglect of its embodied political and religious significances for the state and local society of the Jidu, this paper focuses on the coherent theme of Jidu sacrifice and surrounding this theme I draw four major arguments through critical analysis of primary and secondary sources. 1. The Jidu not only tightly associated with controlling water but was also a symbol and mechanism of political legitimacy. 2. Performing the state sacrificial ritual to the Jidu was thought to be an effective means of connecting mortals and gods, or terrestrial and celestial realms. 3. After the Song dynasty the Jidu God was transformed into a regional protector of local society and savior of local people in addition to an official water god. 4. After the Tang dynasty, the Jidu cult interacted with other religious traditions including Daoism, Buddhism, and popular religion. These arguments and the documentation and demonstration of supporting them present sophisticated and comprehensive research design, questions, and hypotheses.
Point 4: He or she basically says the previous three authors who have written on the Jidu did not cover certain aspects without explain what their works were actually about. The author then embarks on his/her narrative and presentation of findings. This is also an important part of the contextualization which I feel is missing.
Response: In fact, the three authors did only study the historical development of the Jidu sacrifice. This is their main argument. The reviewer could read their papers to see whether there are any other important contributions that I missed.
Point 5: For the English, if you are going to do the copyediting for them (or have someone else do so), it should be no issue. Otherwise, the author should get the help of another native expert to read through the manuscript or, at least, he or she should tell the existing one that there is no need to retain mistakes and stylistic issues to reflect the style of the author. I spotted various grammatical mistakes and stylistic issues in the revised version too. The person helping the author(s) needs to do a more consistent job of helping him/her/them clean up these mistakes. The new version is certainly better than the earlier one. It is clear that the Cambridge professors whom they mentioned in the responses were not reading the paper for language.
Response: Both the previous and revised versions have been reviewed by two renowned UK professors in both content and English language, and a famous Chinese scholar has also helped check the translation and interpretation of cited primary sources for two times. Again, I hope that the reviewer could indicate any concrete errors so that I am able to correct them, not just say “I spotted various grammatical mistakes and stylistic issues”.